# Study Protocol for the Evaluation of “SuperFIT”, a Multicomponent Nutrition and Physical Activity Intervention Approach for Preschools and Families

**DOI:** 10.3390/ijerph17020603

**Published:** 2020-01-17

**Authors:** Ilona van de Kolk, Sanne M. P. L. Gerards, Lisa S. E. Harms, Stef P. J. Kremers, Angela M. H. S. van Dinther-Erkens, Monique Snellings, Jessica S. Gubbels

**Affiliations:** 1Department of Health Promotion, School of Nutrition and Translational Research in Metabolism (NUTRIM), Maastricht University, 6211 MD Maastricht, The Netherlands; sanne.gerards@maastrichtuniversity.nl (S.M.P.L.G.); lisa.harms@maastrichtuniversity.nl (L.S.E.H.); s.kremers@maastrichtuniversity.nl (S.P.J.K.); jessica.gubbels@maastrichtuniversity.nl (J.S.G.); 2Ecsplore, 6160 AC Geleen, The Netherlands; aerkens@ecsplore.nl; 3Spelenderwijs, 6160 AC Geleen, The Netherlands; msnellings@spelenderwijs.nl

**Keywords:** BMI z-score, childcare, environment, family, integrated intervention, nutrition, overweight prevention, physical activity, sedentary behaviour, toddlers

## Abstract

The promotion of healthy energy balance-related behaviours (EBRB) is already important for children at a young age. Different settings, for example childcare and home, play an important role in the EBRB of young children. Further, factors in different types of environment (e.g., physical, sociocultural and political) influence their behaviours. SuperFIT (Systems of Underprivileged Preschoolers in their home and preschool EnviRonment: Family Intervention Trial) is a comprehensive, integrated intervention approach for 2–4 year old children. This paper describes the development and design of the evaluation of SuperFIT. The SuperFIT intervention approach consists of preschool-based, family-based, and community-based components. Intervention activities aimed at changing the physical, sociocultural and political environments in each setting and establishing an increased alignment between the settings. A quasi-experimental design was adopted with twelve intervention and nine control preschools to evaluate effectiveness. The primary outcomes were Body Mass Index (BMI) z-scores (objectively assessed height and weight), dietary intake (24 h recall), and physical activity (accelerometer) of the children. Further, the effects on the nutrition- and physical activity-related practices of preschool teachers and parents were evaluated (questionnaires). Intervention effectiveness was evaluated using linear mixed models. Process evaluation was performed using mixed methods; both quantitative (questionnaires) and qualitative (observations and in-depth interviews) measures were used. The comprehensive, integrated approach of SuperFIT is expected to support healthy EBRB in young children.

## 1. Introduction

Childhood overweight and obesity remain an important public health problem, with a continued expected rise in prevalence in the coming years [1]. In the Netherlands, around 8% of 2 year olds are overweight, and this increases for 4 year olds to 9.1% for boys and 16.3% for girls [2]. It is known that childhood overweight and obesity are likely to track into adulthood [3]. Furthermore, changes in weight status between the age of 2 and 6 years appear to be most predictive for adult overweight [4].

Overweight and obesity are associated with chronic diseases such as diabetes type 2 and cardiovascular diseases, and psychosocial problems that can occur already during childhood [5,6]. This is predominantly the result of unfavourable energy balance-related behaviours (EBRB), such as a high intake of energy-dense food and drinks, low levels of physical activity (PA), and high levels of sedentary behaviour (SB) [7]. Family socioeconomic status and neighbourhood deprivation are important determinants of overweight and obesity [8,9]. In order to prevent childhood overweight and obesity, the promotion of healthy nutrition and PA in young children is essential, particularly in high-risk groups [10,11].

SuperFIT was developed as a comprehensive, integrated intervention approach to promote healthy EBRB in young children (2–4 years old). It is based on three main principles. The first one involves a combined focus on nutrition and PA. Childhood obesity, as well as EBRB, are often the result of a complex interplay between nutrition and PA behaviour [7]. Furthermore, unhealthy nutrition and PA habits often cluster within the same children [12,13]. Young children are often highly sedentary, with limited physical activity [14,15,16]. They already show unhealthy dietary patterns, and even those with healthier dietary patterns do not always comply with nutritional guidelines [17,18]. In general, adherence to dietary guidelines is low, especially for vegetable and fruit intake, sugar intake and total energy intake [19,20,21]. Therefore, SuperFIT primarily focusses on increasing fruit and vegetable consumption, decreasing unhealthy snack consumption, increasing water consumption, increasing PA, and decreasing sedentary behaviour.

The second principle highlights the multi-setting approach, as it targets the childcare, home, and community settings. Children’s EBRB are influenced within different (micro-)systems [22]. In countries belonging to the Organization for Economic Co-operation and Development (OECD), a majority of young children are partially cared for in formal childcare [23]. The childcare setting is therefore an important micro-setting related to children’s EBRB, along with the home setting [24,25,26,27]. From a systems perspective, it is important to ensure the alignment of these different micro-settings, in order to induce synergetic effects [28,29].

The childcare setting is regarded as promising for the implementation of interventions to promote healthy child EBRB [10]. Evaluations of these interventions have shown their potential to affect EBRB and weight-related outcomes positively [30,31,32,33]. Interventions have been implemented in the home setting with positive effects on the children’s EBRB [34,35]. While the integration of childcare and home settings has been increasingly recognized as supporting intervention effectiveness [33,36], the results of integrating childcare-based and family-based interventions have been inconclusive [34,37]. This may mainly be attributed to the type of parental involvement, with direct (or active) involvement (e.g., parents’ attendance at training or educational sessions) being more supportive of changes in their children’s behaviour [36]. The intensity of parental involvement may be influential, with more intensity being supportive of intervention effectiveness [36].

The inclusion of a community setting has also been shown to be supportive of the prevention of childhood overweight and obesity, particularly when combined in a multi-setting approach [38]. The community can contribute by, for example, increasing access to PA opportunities [39]. Establishing connections and cooperation with community partners increases the sustainability of changes [40].

The third principle focuses on the integration of different types of environments in the SuperFIT approach. Socio-ecological models underline the influence of determinants of EBRB in the environment [22,41]. Crucial determinants of excessive weight gain in toddlers can be identified in the sociocultural environment (e.g., parenting style and nutritional and PA-related parenting practices [42,43,44]), physical environment (e.g., availability of play materials, play space and healthy food products [15,45,46]), economic environment (e.g., costs of food products [47,48]), and political environment (e.g., formulating clear policies in childcare [49]) [50]. In addition, socio-ecological models suggest an interaction between these different types of environments [22,29]. For example, the effects of changes in the physical environment may be moderated by changes in the sociocultural environment, and similarly for any other combination of environments [29]. Therefore, it is important to take into account the different types of environments in intervention development. The SuperFIT approach aims to integrate changes in specifically the physical, sociocultural and political environments, because they are the most changeable types of environments within the three settings (childcare, home, community) [50].

The SuperFIT approach assumes that the incorporation of these principles within intervention strategies will result in greater effects to prevent overweight and obesity in young children [29]. The childcare setting is considered the primary one, particularly due to its possible point of entry. The SuperFIT approach is not a pre-specified intervention programme, but is adaptable to the individual situations of childcare organizations. Intervention efforts are therefore focused on the aspects requiring change. The aim of the SuperFIT approach is to improve the EBRB of 2–4 year old children and prevent overweight and obesity. It is expected that the SuperFIT approach will increase physical activity and decrease sedentary behaviour. Further, it is expected that the SuperFIT approach will increase the intake of fruit and vegetables, and water, and decrease the intake of unhealthy snacks and sugar-sweetened beverages. For BMI z-score, it is expected that the SuperFIT approach will help young children to maintain or achieve a healthy BMI z-score.

The SuperFIT approach, as implemented in a pilot region, will be evaluated through an effect and process evaluation. The aim of the effect evaluation is to assess its effectiveness on the BMI z-score, PA, sedentary behaviour and dietary intake (primary outcomes) of children aged 2–4 years old from disadvantaged families in the Netherlands. In order to do so, a quasi-experimental design will be adopted, and the intervention group will be compared to a control group that does not receive the SuperFIT intervention approach. Changes in the sociocultural environment (i.e., nutritional and PA-related practices of preschool teachers and parents) and the physical environment will also be assessed. The aim of the process evaluation will be to gain insight into the processes supporting the development and implementation of SuperFIT. This will be used to better understand the results of the effect evaluation and support their interpretation. The current paper describes the content of the SuperFIT approach in the pilot region and the research protocol concerning the evaluation.

## 2. Materials and Methods

### 2.1. Study Design

For the evaluation of SuperFIT, the RE-AIM (Reach, Effectiveness, Adoption, Implementation, and Maintenance) framework will be used as a guide [51]. A mixed methods design will be used for the evaluation. In order to assess the effectiveness of SuperFIT, a quasi-experimental research design will be adopted. In addition, process evaluation using qualitative and quantitative research methods will be performed to evaluate Reach, Adoption, Implementation and Maintenance, in addition to Effectiveness. The SPIRIT (Standard Protocol Items: Recommendations for Intervention Trials) was used as guideline to draft the study protocol (see Appendix A: SPIRIT checklist) [52].

### 2.2. Study Setting

In the Netherlands, formal centre-based childcare takes two forms. First, day-care centres provide full-day childcare [53], which children aged 0–4 years old can attend. Second, preschools provide half-day childcare with the specific goal to prepare 2–4-year-old children in a playful way for primary school [53]. Parents can receive a general childcare benefit for formal childcare from the government, based on their working hours and income [54]. The SuperFIT approach was implemented at *preschools* in the pilot region because they have a broader reach compared to day-care centres. Children with language or socio-emotional developmental delays, for example, can be referred to preschools to undergo a program to alleviate these delays [55]. This results in the inclusion of vulnerable groups. Day-care centres are mostly used by families with working parents and higher incomes, which would result in a restricted sample [56].

### 2.3. The Intervention

SuperFIT was developed in a partnership with the preschool organization in the pilot region, a local PA-providing organization, and health promotion experts. A steering committee of stakeholders, including the municipality, community health service and youth health care agency, were consulted during the process of development and implementation. As formative research for the intervention development, a needs assessment was performed among preschool teachers and parents of the target population [48]. The theory and evidence-based knowledge from the health promotion experts, practice-based knowledge of the partners, and the input of the formative research were used to develop the different intervention components and strategies. During the implementation, a continuous process of co-creation, feedback and adaptations was adopted to develop the SuperFIT approach. The focus was to select strategies that could be considered add-in as opposed to add-ons. In other words, SuperFIT was designed to be integrated into daily routines as much as possible, rather than demanding additional activities to daily routines (e.g., additional physical education classes). Furthermore, intervention strategies were developed in such a way that there was a high adaptability to the specific situation of preschool teachers in their daily work.

#### 2.3.1. Preschool-Based Component

The preschool-based component of SuperFIT aimed at changes in its sociocultural, physical and political environments. The sociocultural environment was operationalised as the nutritional and PA-related practices of the preschool teachers. Different strategies were applied to promote healthy practices. First, an inspirational session was organized for the preschool teachers with a well-known Dutch professional in the field of school-based PA. Second, three 2 h, off-the-job training sessions were provided for the preschool teachers [57]. All training sessions consisted of three sub-sessions led by an expert on the following topics: PA and related practices at the preschool; nutrition and related practices at the preschool; and positive child-rearing style. Preschool teachers got the opportunity to choose which of the sub-sessions they would attend based on their personal learning goals. At least one teacher of each preschool was expected to attend each sub-session, so that all themes would be covered within one preschool. The sessions were highly interactive and promoted an exchange of experiences between the attendees. Third, an on-the-job coaching session was provided by a PA and health coach after all off-the-job training [58]. Lastly, to support the preschool teachers at the workplace, PA and nutrition cards were developed. They contain easy-to-perform PA games and nutrition-related activities that fit with the current learning methods used in the preschools. They were developed by the experts within the SuperFIT partnership.

The physical environment was defined as ‘what is available at the preschool’ [50]. For PA, the strategies focused on increasing the availability of play materials. All preschools received a box with general PA-promoting play materials. These materials were aligned with the PA cards to enable all preschools to perform the activities described on the cards. The box contained a variety of materials that could promote PA both indoors and outdoors, such as bean bags, hoops, balls, sidewalk chalk, and clothespins [59]. In addition, an assessment of preschool-specific needs for materials was performed in order to provide these additional materials (e.g., stepping-stones or foam blocks).

Regarding nutrition, the strategies focused on increasing the variety and availability of fruit and vegetables during snack time and providing general nutrition-related materials [17]. A local greengrocer supplied unfamiliar fruits or vegetables (e.g., cherries, raspberries, avocado, celery) to increase the variety of fruits and vegetables. This supplemented the fruits that the children would bring to the preschool from home and was available every day. The supplied fruits or vegetables were similar during the two weeks to increase repeated exposure of the children to each new product [45,60]. The general nutrition-related materials were part of the general box, with play materials, and were matched with nutrition-related cards. Materials included a water tap, fruit and vegetable toys, nutrition-related story books, and materials to involve children in preparing foods. Preschool teachers could also express the need for specific nutrition-related materials (e.g., a blender) to supplement the general materials that were delivered.

The political environment was defined as ‘the institutional policies related to nutrition and PA’ [50]. The strategies of SuperFIT focused on updating the nutrition policy and initiating the development of a PA policy, as this was not yet in place. Particular subjects of interest for the nutrition policy were the availability of water and healthy treats and preschool teacher practices. The PA policy was formulated to provide recommendations on the amount of time at childcare that should be spent active. It was intended to provide guidelines around safe play, particularly in a physical education room.

#### 2.3.2. Family-Based Component

For families of the children in the participating preschools, a family-based component was developed within the SuperFIT partnership. The formative research was used as a guide, but parents were not actively involved in its development. The aim was to use fun family activities to help families integrate healthy nutrition and PA into their normal life. Fathers, mothers, siblings, grandparents, uncles and aunts were all welcome to join. The family sessions were characterised by fun activities for the whole family that concerned PA and nutrition. This included, for example, activity games that could be easily translated to the home setting, tasting sessions of new fruits and vegetables, and making healthy treats.

To be able to address the influences of the different types of environments on nutrition and PA (e.g., nutritional and PA-related parenting practices, availability of (un)healthy food products, rules around screen time), caregiver-only sessions were held in addition to the family sessions. Lifestyle Triple P seminars [61] were given by a trained Triple P provider. The sessions were highly interactive, enabling caregivers to share their experiences, solutions and ideas. Three 1.5 h caregiver-only sessions were provided.

Three rounds of the family-based component were organized. In the first round, four, one hour family sessions were organized. Together with the caregiver-only sessions, a total of seven sessions were delivered. However, parents and implementers indicated that this was too demanding. Therefore, in the second and third rounds, a total of five sessions were organized, three caregiver-only sessions and two family sessions. During the caregiver-only sessions, the implementers organized activities relating to PA and nutrition for the children. For younger siblings who were unable to participate in the sessions, childcare was available.

#### 2.3.3. Community Component

The community component was based on PA and healthy nutrition initiatives that were already available in the intervention region. The aim was to improve linkages between different organisations and increase publicity about PA opportunities available within the community. Therefore, a social map showing sports organizations, playgrounds and a petting zoo was developed and distributed within the community.

### 2.4. Planning

The intervention activities of the preschool component started in April 2017. The first off-the-job training took place in May 2017, followed by an on-the-job coaching. The second and third off-the-job trainings took place after the summer holidays in September 2017, each training followed by an on-the-job coaching. The box with general play materials was available for the preschools after the first off-the-job training. The delivery of supplementary fruits and vegetables was started in May 2017 and lasted until May 2018. The first round of the family-based component started in May 2017, the second round in September 2017, and the third round in January 2018.

### 2.5. Participants

A convenience sample of intervention preschools was recruited from a childcare organization in an urban municipality in Limburg (the Netherlands), based on the socio-economic status (SES) of their neighbourhood. Preschools could participate if they were located in the low-SES neighbourhoods of the pilot region. SES was based on the 2014 values of the Netherlands Institute for Social Research (SCP), with a negative score indicating a low SES [62]. Together with the management of the childcare organization, eligible preschools were selected. No other inclusion or exclusion criteria were applied. In total, twelve preschools participated in SuperFIT. Control preschools were selected in another urban area in Limburg in the south of the Netherlands. This area was comparable with regard to SES. One childcare organization collaborated, and a total of nine preschools participated as a control group. Due to the nature of the project, no randomization was performed. The Maastricht University Medical Centre, Medical Ethics Committee reviewed and approved this study (METC163022/ NL 58061.068.16), and the trial was prospectively registered (Clinicaltrials.gov, NCT03021980).

Children attending the participating preschools were eligible for inclusion. Additional inclusion criteria were: (1) at least one parent had to be able to understand Dutch, and (2) both parents signed the informed consent. Written information about the SuperFIT project was sent to each preschool to hand out to all parents with children attending that preschool. This information leaflet also informed the parents of the family-based component. Two weeks later, a researcher visited the preschool to explain SuperFIT verbally, starting with a kick-off event organised by the SuperFIT partnership. During that time, the parents were able to ask for additional information and hand in their informed consent for participation in the preschool-based component research and, additionally, the family-based component.

During the course of SuperFIT, additional recruitment efforts were made to increase participation in the family-based component. First, parents of the participating preschools were informed through newsletters of the new rounds starting the family-based components. Second, parents in other preschools in the pilot region were informed about the family-based component and invited to participate.

All preschool teachers working at the participating preschools were part of the target population of SuperFIT as intermediaries for child EBRB. They were informed about SuperFIT by the childcare organisation during the development phase. Written information about SuperFIT was also sent to them. Two weeks later, a researcher visited the preschool to explain SuperFIT verbally, and the preschool teachers were able to provide their informed consent at that time.

## 3. Data Collection

For the effect evaluation, baseline measurements were performed before the start of the intervention, from January until April 2017. In the control group, baseline measurements were performed from January until July 2017. Follow-up measurements took place in November/December 2017 (first follow-up) and May/June 2018 (final follow-up). In order to reduce the participant burden, the data collection was aligned with intervention participation. This means that more elaborate data were collected for participants in the family-based component compared to participants in the preschool-based component or control group.

For the process evaluation, data were collected continuously during the implementation period. Qualitative and quantitative data will be entered, cleaned, coded and analysed from July 2018 until July 2020. Figure 1 shows the planning of the research and implementation of SuperFIT.

### 3.1. Effect Evaluation

#### 3.1.1. BMI z-Score

A trained member of the research team will assess the weight, height and waist circumference of the children, using a standardized protocol. Standing height will be measured to the nearest decimal in centimetres (cm) using the Seca© 213 stadiometer (Seca, Hamburg, Germany), with light clothing and without shoes. Weight will be measured to the nearest decimal in kilograms (kg) using the Seca© Clara 803 (Seca, Hamburg, Germany), digital weighing scale. Heavy clothing and shoes will be removed before measurement. Waist circumference will be measured using the Seca© 201 (Seca, Hamburg, Germany), measuring tape. Only a thin vest or t-shirt between the measuring tape and skin will be allowed. A single measurement will be performed to assess height, weight and waist circumference. Anything unusual occurring during the measurements, such as not wanting to take off shoes or wearing heavy clothing, will be recorded to adjust for it during analysis. Height and weight measurements will be used to calculate BMI, which will be converted to a BMI z-score, adjusted for age and gender using a Dutch reference population (the Fifth Growth Study) [2].

Similar anthropometric measurements will be performed on one of the parents of each family participating in the family-based component, using the same protocol and measurement instruments. Weight and height measurements will be used to calculate BMI.

#### 3.1.2. Dietary Intake

The children’s dietary intake will be measured both at home and at the preschool. A 24 h dietary telephone recall will be conducted to assess dietary intake at home. Researchers and research assistants will be trained in following a dietary recall protocol and entering data in the Blaise© software (version 4.8.4.1767 (Statistics Netherlands (CBS), The Hague, the Netherlands)). Phone calls will be done in the evening adopting a structured protocol that divides a day into seven chronological eating moments: yesterday’s evening snack(s), today’s breakfast, morning snack(s), lunch, afternoon snack(s), dinner and evening snack(s). Parents will be asked to report food products consumed, starting with yesterday’s evening snack(s). Fruit, vegetables and snacks will be the major focus of the dietary recall. This meant that snacking moments were explored in detail, while the questions about main meals focused on fruits, vegetables and beverages only.

Details of each product will be requested, such as the kind, portion size, amount and preparation technique. Probing questions (e.g., did he/she drink something during dinner? Was this regular soda or diet soda?) will be used as memory cues to help parents record all products, including product details.

Blaise© (version 4.8.4.1767 (Statistics Netherlands (CBS), The Hague, the Netherlands)), a system used to administer computer-controlled questionnaires, will serve as the data entry software and contain child products (e.g., candy, fruit drinks) and child portion sizes (e.g., segments of fruit and a sippy cup) to match a child’s diet. It also will provide an ‘unknown’ option, an ‘other’ option and a comment section whenever the existing codes will not match. These will be recoded into existing or new product codes later in the process. Blaise was connected to The Dutch Food Composition Database, version 2016/5.0 (National Institute for Public Health and the Environment (RIVM), Bilthoven, the Netherlands) to assess nutrient composition.

At the preschool, the children’s dietary intake will be assessed using a dietary journal [17]. The teachers will record the intake of each child on a predefined list of the most commonly consumed food products and beverages at the preschool. Additional blank spaces will be available for any other food products consumed. Consumption will be recorded in number of units most common for the food product (e.g., parts for fruits/vegetables, pieces for sweets, cups for beverages).

For families participating in the family-based component, a short food frequency questionnaire on fruits/vegetables, sweets and snacks, and beverages will be part of the measurement diary, which will be provided with the accelerometer, to assess parental dietary intake.

#### 3.1.3. Physical Activity and Sedentary Behaviour

Children’s PA and sedentary behavior (SB) will be assessed using Actigraph GT3X+ (Actigraph, Pensacola, FL, USA) accelerometers, applying an adjusted wearing protocol [63]. Accelerometers will be placed on the right hip using an elastic belt. Children will wear the accelerometers for eight consecutive days during waking hours. Instructions will be given to remove the accelerometer for activities involving water such as bathing, showering and swimming. Parents will be provided with a measurements diary to record wear-time particularities, preschool attendance, and attendance at other childcare facilities.

PA and SB of one parent of the families participating in the family-based component will be assessed using Actigraph GT3X+ (Actigraph, Pensacola, FL, USA) accelerometers. The measurement protocol for the children also will apply to their parents.

#### 3.1.4. Questionnaires

The preschool teachers will be asked to complete a questionnaire on demographic variables and nutritional and PA-related practices (Child-care Food and Activity Practices Questionnaire, CFAPQ) [64]. All parents will be asked to fill out a questionnaire on demographics and other background variables. The parents of children in the family component will be additionally asked to fill out a questionnaire on nutritional and PA-related practices (Preschooler Physical Activity Parenting Practices questionnaire (PPAPP) [65], and Comprehensive Feeding Practices Questionnaire (CFPQ) [66]), family health climate (Family Health Climate Scale [67]), and physical home environment, based on the Environment and Policy Assessment and Observation-Self Report (EPAO_SR) [68]).

#### 3.1.5. Preschool Physical Environment

Questions from the Environment and Policy Assessment and Observation instrument (EPAO) [69] related to the physical environment will be adapted to the Dutch setting and will be used to assess the physical preschool environment. A trained researcher will observe each location and will fill out the questionnaire at baseline and both follow-up measurements.

### 3.2. Process Evaluation

The process evaluation will be conducted to gain insight into the reach, adoption, implementation and maintenance of SuperFIT, using both quantitative and qualitative measurements. Preschool teachers and parents in the intervention group will be asked questions about their appreciation of SuperFIT in the follow-up questionnaires. Observations will be done at the preschool locations to assess implementation fidelity, change in daily activities, and the social and physical environment at the preschool. The observations were done in one morning, twice during implementation (September/October 2017 and April 2018) and once after implementation (September 2018).

In-depth semi-structured interviews were performed with the preschool teachers, parents, management, and implementers on several occasions during or following implementation. In June and July 2017, in-depth interviews were held with the preschool teachers, focusing on development and implementation. These interviews were also used to adapt the intervention strategies that were still to come.

In February and March 2018, in-depth interviews with the preschool teachers were held to gain insight into their experiences with SuperFIT, such as its strengths and limitations, and the facilitators and barriers of integrating SuperFIT into daily practice. Finally, in October and November 2018, in-depth interviews were held with preschool teachers, management and implementers that focused on the maintenance of SuperFIT within their organisation. After each round of the family-based component, in-depth interviews were held with the participating parents on their experiences, strengths and limitations, and changes that may have occurred as a result of their participation.

All intervention activities were observed using a free-form protocol to record any aspects occurring during the activities and give a general impression of the intervention activity. Attendance at the intervention activities was recorded in order to evaluate reach.

### 3.3. Data Analysis

Quantitative continuous variables will be presented as means and standard deviations. Categorical data will be presented by percentages of participants in each of the possible categories. Baseline characteristics and outcome values will be analysed for differences between the groups, using analysis of variance (ANOVA) for continuous variables and chi-square tests for categorical variables. The effects of SuperFIT will be analysed using linear mixed models with child and preschool levels in order to correct for repeated measurements and group effects. Known potentially relevant confounders will be taken into account based on the literature and/or differences in baseline characteristics. All analyses will be performed using SPSS version 25.0 (IBM Corp, Armonk, NY, USA).

Qualitative data of the interviews will be audio-recorded and transcribed verbatim. Interview transcripts will be coded by themes and concepts using NVivo version 11 (QSR International, Doncaster, Victoria, Australia).

### 3.4. Sample Size

An a priori sample size calculation was performed based on the BMI z-score. For the preschool-based component, the expected difference between the intervention and control condition was 0.10 BMI points [70]. Given a power of 0.90 and α < 0.05, a sample of 115 children in each group was required to detect this difference. Correcting for the potential nesting of effects within a preschool, considering an intraclass correlation of 0.0006, which corresponds to a design effect of 1.23, a total of 142 children should be included. Taking into account an attrition rate of 20%, 171 children should be included in each study group, resulting in 342 children.

For the family-based component, the expected difference between the intervention and control group is 0.30 BMI z-points [70]. Taking into account a power of 0.90 and α < 0.05, a sample of 38 families is required. Adjusting for an attrition rate of 30% a total of 50 families is the target for inclusion in the family-based component.

Recruitment for this study was done between January and April 2017 for the intervention group and between January and July 2017 for the control group. A flow diagram of participation is shown in Figure 2. Parents of 23.9% of the children attending intervention preschools agreed to participate in the preschool component, 41.0% of these parents also agreed to participate in the family-based component, and 26.7% of parents of children in the control preschools agreed to participate. Of the preschool teachers, 91.4% from the intervention preschools and 65.0% from the control preschools agreed to participate.

As the a priori calculated sample size was not reached, an additional power calculation was conducted. Based on the sample size (intervention *N* = 99; control *N* = 92), a difference of 0.19 BMI z- points can be detected, which seems attainable based on the available evidence [37,70]. The calculations are based on the same assumptions of the a priori sample size calculation. In addition, an increase of 1.44% of time in moderate-to-vigorous physical activity (MVPA) per day (corresponding to 8.91 min per day), 0.33 instances of fruit consumption per day, 0.26 instances of vegetable consumption per day, and 0.57 instances of water consumption per day can be detected. A decrease of 0.52 instances of sugar-sweetened beverages consumption per day and 0.61 instances of snack consumption per day can be detected. These differences also seem attainable based on the available evidence [59,71].

## 4. Discussion

This study protocol describes the design of the effect and process evaluation of SuperFIT, a comprehensive, integrated intervention approach. It aims at affecting the children’s EBRB through changes in multiple types of environments and aligning these changes in the preschool, home and community settings. In particular, targeting both the preschool and the home settings may be important for intervention effectiveness, as socio-ecological theories and research describe an interplay between them [28,29,72]. Intervention research has shown that incorporating a parental component in childcare-based interventions may be essential for intervention effectiveness [30,33]. Furthermore, not targeting single behaviours or types of environment in isolation, but rather taking into account the complexity of childhood overweight and obesity was expected to be supportive of effects on children’s EBRB and weight-related outcomes [30].

SuperFIT was developed in close collaboration between practice professionals and health promotion experts, in co-creation with the target group. This enhanced its applicability and usability [73], and a rigid evaluation of the program was ensured [74]. The SuperFIT approach was developed to be adaptable to the specific situation of a childcare organization and location, which may foster sustainability [75]. It also contains elements (e.g., on-the-job coaching) that directly assist childcare workers in its application in the context of their daily practice, and therefore stimulates implementation [76]. The extensive process evaluation will study these factors and try to understand the changes that occur within the system that may or may not lead to effects from the SuperFIT approach [77,78].

The effect evaluation was done using objective measurements where possible (i.e., accelerometer data, height and weight measures), valid measurement of dietary intake (24 h recall), and validated questionnaires (practices). One limitation of the study may be the quasi-experimental design, without randomisation. Convenience samples of the preschools were used, which may have introduced selection bias. However, in intervention research, it is important to find a balance between internal (i.e., rigorous research designs) and external (i.e., generalizability) validity [79]. For the evaluation of the SuperFIT approach, the current study design was considered most appropriate for achieving this balance.

## 5. Conclusions

SuperFIT is a multi-component, integrated intervention that aims to promote healthy EBRBs in young children through aligning the childcare and home settings with regard to physical activity and healthy nutrition. A rigid effect and process evaluation will provide insight on the possible effectiveness of this type of intervention and factors that may have influenced this effectiveness.

## Figures and Tables

**Figure 1 ijerph-17-00603-f001:**
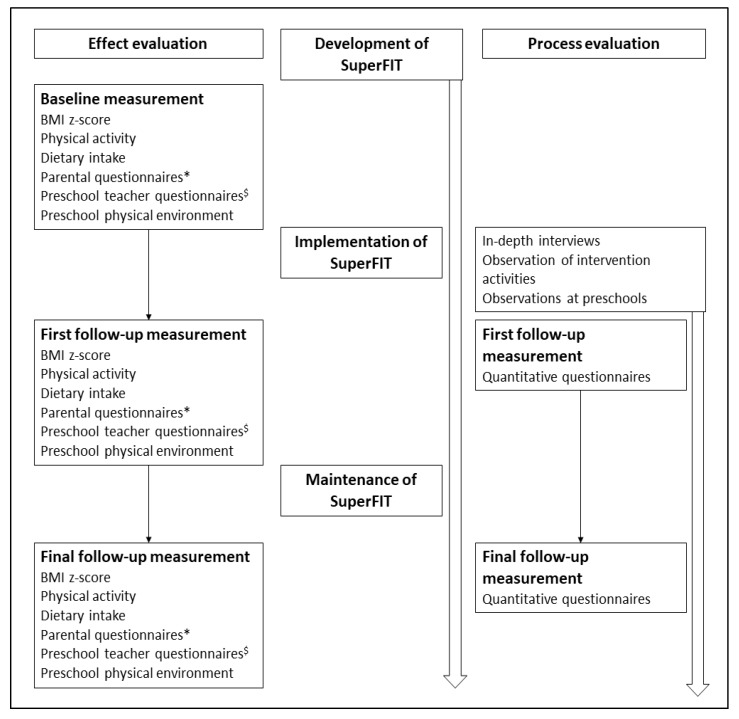
Planning of the implementation and evaluation of SuperFIT (Systems of Underprivileged Preschoolers in their home and preschool EnviRonment: Family Intervention Trial). * Parental questionnaires measured demographics, nutritional and physical-activity-related practices, family health climate and physical home environment. ^$^ Preschool teacher questionnaires measured demographics and nutritional and physical-activity-related practices.

**Figure 2 ijerph-17-00603-f002:**
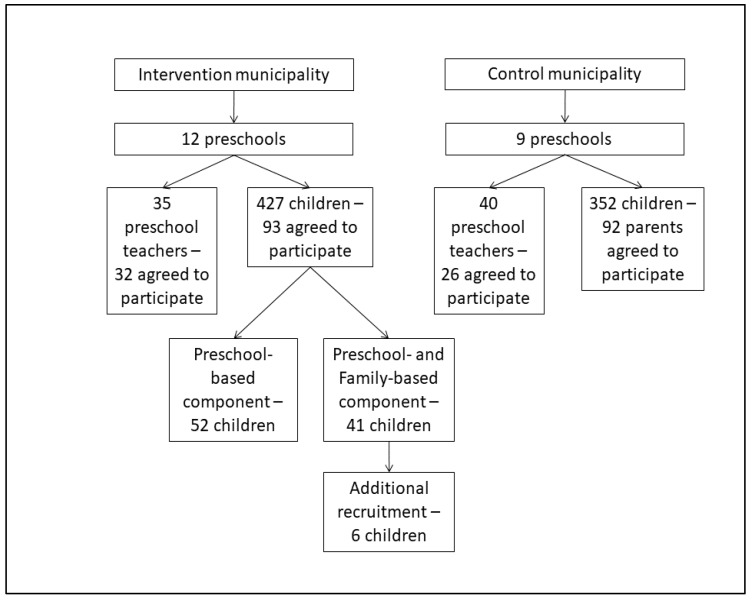
Flow diagram of the participants of SuperFIT.

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
