# Peer review of "Study Protocol for the Evaluation of “SuperFIT”, a Multicomponent Nutrition and Physical Activity Intervention Approach for Preschools and Families"

_ijerph, 2020, doi:10.3390/ijerph17020603_

Round 1
Reviewer 1 Report
This manuscript (protocol) deals with a nutrition-related issue which is on the WHO agenda of important health topics deserving evidence-based interventions. In this study protocol, the authors provide a comprehensive and integrated program named “SuperFIT” which includes both interventions and process evaluation. The aims of this study are very interesting and helpful, and the results may facilitate implementing widescale health promotion strategies among preschoolers. From my point of view, this article is written in a comprehensive manner and deals with all significant points.
For enhancing the information regarding the necessity of nutrition interventions, I invite the authors to read and consider relevant papers such as:
https://www.who.int/nutrition/publications/essential-nutrition-actions-2019/en/
Author Response
This manuscript (protocol) deals with a nutrition-related issue which is on the WHO agenda of important health topics deserving evidence-based interventions. In this study protocol, the authors provide a comprehensive and integrated program named “SuperFIT” which includes both interventions and process evaluation. The aims of this study are very interesting and helpful, and the results may facilitate implementing widescale health promotion strategies among preschoolers. From my point of view, this article is written in a comprehensive manner and deals with all significant points.
For enhancing the information regarding the necessity of nutrition interventions, I invite the authors to read and consider relevant papers such as:
https://www.who.int/nutrition/publications/essential-nutrition-actions-2019/en/
Author response:
Thank you for the compliments on our study and our protocol. As the statement about the importance of health promoting interventions was already made in the introduction, we have added the suggested reference to that statement. In addition, we have added a reference to support this statement with regard to physical activity.
Changes:
Introduction, line 50: In order to prevent childhood overweight and obesity, the promotion of healthy nutrition and PA in young children is essential, particularly in high-risk groups [10,11].
Reviewer 2 Report
This manuscript is highly interesting. This intervention (SuperFIT) consists of a preschool based, family-based, and community-based component. Intervention activities aimed at changing the physical, socio-cultural and political environments in each setting and establishing increased alignment between the setting. However, there is something that the authors have to review:
- First, they sometimes write in the present tense and other times in the future tense. Given that this paper is a protocol, I think it must be written in the future tense.
- The most important, the authors should know the SPIRIT check-list: explanation and elaboration: guidance for protocols of clinical trials. It is necessary the authors take into account this check-list and proving it in this paper. As well, they must add to the paper a Table or Appendix with this check-list answering each item one and one. On the other hand, the authors also have to take into account check-list for quasi-experimental studies (non-randomized experimental studies) and to add another Table or Appendix proving (explain it) the items one by one on this study.
This checking will help the authors to test its own manuscript.
Author Response
This manuscript is highly interesting. This intervention (SuperFIT) consists of a preschool based, family-based, and community-based component. Intervention activities aimed at changing the physical, socio-cultural and political environments in each setting and establishing increased alignment between the setting. However, there is something that the authors have to review:
- First, they sometimes write in the present tense and other times in the future tense. Given that this paper is a protocol, I think it must be written in the future tense.
Author response:
Thank you for your compliments and remarks. We have checked the manuscript and where appropriate we have changed the tense used. However, as some aspects described in the protocol took place in the past, the future tense is not always appropriate.
Changes:
Throughout the manuscript.
- The most important, the authors should know the SPIRIT check-list: explanation and elaboration: guidance for protocols of clinical trials. It is necessary the authors take into account this check-list and proving it in this paper. As well, they must add to the paper a Table or Appendix with this check-list answering each item one and one. On the other hand, the authors also have to take into account check-list for quasi-experimental studies (non-randomized experimental studies) and to add another Table or Appendix proving (explain it) the items one by one on this study.
This checking will help the authors to test its own manuscript.
Author’s response:
We are indeed familiar with the SPIRIT-checklist and this was used during drafting of the manuscript. We have now also provided it in the appendix of the manuscript.
In addition, we used the TREND (Transparent Reporting of Evaluations with Non-randomized Designs) checklist regarding the quasi-experimental design of the study to check the manuscript. However, as this checklist applies to the reporting of quasi-experimental trials, not protocols, not everything is applicable for this manuscript. Therefore, we chose not to report it in the appendix of this manuscript. To comply with the SPIRIT checklist, some changes are made to the manuscript.
Changes:
To comply with checklist item 7:
Introduction, line 103-108: The aim of the SuperFIT approach is to improve the EBRB’s of 2-4 year old children and prevent overweight and obesity. It is expected that the SuperFIT approach will increase physical activity and decrease sedentary behaviour. Further, it is expected that the SuperFIT approach will increase the intake of fruit and vegetables, and water, and decrease the intake of unhealthy snacks and sugar sweetened beverages. For BMI z-score it is expected that the SuperFIT approach will help young children to maintain or achieve a healthy BMI z-score.
To comply with checklist item 6b and 8:
Introduction, line 112-114: In order to do so, a quasi-experimental design will be adopted and the intervention group will be compared to a control group that does not receive the SuperFIT intervention approach.
Appendix: Spirit checklist
Reviewer 3 Report
Good idea. Please define: OECD.
I recommend a table for the methods too so its easier to understand. The writing is to dense to read and hard to interpret if not familial with study.
Are there any results so far?
Author Response
Good idea. Please define: OECD.
Author’s response:
Thank you for the compliment. OECD stands for Organization for Economic Co-operation and Development. We have added this also to the manuscript.
Changes:
Introduction, line 64: In countries belonging to the Organization for Economic Co-operation and Development (OECD), a majority of young children are partially cared for in formal childcare [21].
I recommend a table for the methods too so its easier to understand. The writing is to dense to read and hard to interpret if not familial with study.
Author’s response:
Thank you for your suggestion. We have added a figure to the methods section showing the study design to support the text. In addition, we revised the methods section for clarity.
Changes:
Methods, line 274: Figure 1 shows the planning of the research and implementation of SuperFIT.
Figure 1: Figure 1. Planning of the implementation and evaluation of SuperFIT. *Parental questionnaires measured demographics, nutritional and physical activity-related practices, family health climate and physical home environment. $Preschool teacher questionnaires measured demographics and nutritional and physical activity related-practices.
-- FIGURE UPLOADED AS SEPARATE FILE AND IN THE MANUSCRIPT --
Are there any results so far?
Author’s response:
The results of this study will be published separately. There are already results available on the BMI-z score and physical activity outcomes here: https://www.mdpi.com/1660-4601/16/24/5016. The manuscript describing the results on the nutrition outcome is currently under review. Analysis of the process evaluation is still ongoing.
Round 2
Reviewer 2 Report
The authors have replied very well to different comments. In my opinion, the manuscript is ready for publishing.